# PlacidDreamer: Advancing Harmony in Text-to-3D Generation

## ABSTRACT

Recently, text-to-3D generation has attracted significant attention, resulting in notable performance enhancements. Previous methods utilize end-to-end 3D generation models for initializing 3D Gaussians, and multi-view diffusion models to enforce multi-view consistency. Moreover, they employ text-to-image diffusion models to refine details with score distillation algorithms. However, these methods exhibit two limitations. Firstly, they encounter conflicts in generation directions since different models aim to produce diverse 3D assets. Secondly, the issue of over-saturation in score distillation has not been thoroughly investigated and solved. To address these limitations, we propose PlacidDreamer, a text-to-3D framework that harmonizes initialization, multi-view generation, and text-conditioned generation with a single multi-view diffusion model, while simultaneously employing a novel score distillation algorithm to achieve balanced saturation. To unify the generation direction, we introduce the Latent-Plane module, a training-friendly plug-in extension that enables multi-view diffusion models to provide fast geometry reconstruction for initialization and enhanced multi-view images to personalize the text-to-image diffusion model. To address the over-saturation problem, we propose to view score distillation as a multi-objective optimization problem and introduce the Balanced Score Distillation algorithm, which offers a Pareto Optimal solution that achieves both rich details and balanced saturation. Extensive experiments validate the outstanding capabilities of our PlacidDreamer. The code will be available on GitHub.

## CCS CONCEPTS

• **Computing methodologies** → **Computer vision problems**.

## KEYWORDS

3D Generation, text-to-3D, score distillation

## 1 INTRODUCTION

The task of generating 3D assets from text, known as text-to-3D, has garnered significant attention for its potential to simplify 3D creation, a process once requiring specialized knowledge. Due to the relative scarcity and lower quality of 3D data compared with 2D data, one promising approach is to adapt pre-trained 2D models for 3D generation. An optimization-based approach leveraging score distillation algorithms, which distill generative capability from pre-trained 2D diffusion models to guide subsequent 3D generations, has emerged as a dominant approach in this field.

Permission to make digital or hard copies of all or part of this work for personal or classroom use is granted without fee provided that copies are not made or distributed for profit or commercial advantage and that copies bear this notice and the full citation on the first page. Copyrights for components of this work owned by others than the author(s) must be honored. Abstracting with credit is permitted. To copy otherwise, or republish, to post on servers or to redistribute to lists, requires prior specific permission and/or a fee. Request permissions from permissions@acm.org.
ACM MM, 2024, Melbourne, Australia
© 2024 Copyright held by the owner/author(s). Publication rights licensed to ACM.
ACM ISBN 978-x-xxxx-xxxx-x/YY/MM
https://doi.org/10.1145/nnnnnnn.nnnnnnn

Since the introduction of the first score distillation algorithm, Score Distillation Sampling (SDS) [37], subsequent works have significantly advanced this optimization-based approach, enhancing both generation quality and speed. One significant factor influencing generation quality is the multi-face problem. To address this problem, Magic123 [38], DreamCraft3D [47], Consistent123 [27], and EfficientDreamer [61] incorporate multi-view diffusion models to enhance multi-view consistency. More recently, the introduction of 3D Gaussian Splatting [22] has further optimized the pipeline with convenient initialization, faster rendering, and training speed. LucidDreamer [25], GaussianDreamer [58], and GSGEN [6] propose to leverage end-to-end 3D generation models [20, 35] to provide a robust 3D Gaussian initialization, thereby enhancing overall quality.

Despite significant progress in the aforementioned methods, there are still two main limitations that require attention:

- **Conflicting Optimization Directions.** The integration of multiple generative models within a single pipeline can lead to contradictory optimization directions. For instance, the score distillation guidance derived from multi-view diffusion models may be at odds with that from text-to-image diffusion models [38], necessitating the development of specialized balancing strategies [27] or additional refinement [47] for optimal performance. Moreover, employing distinct generative models for supervision across different stages [6, 58] can cause the generative model in later stages to ignore the outputs of its predecessors and independently generate new results based on its intrinsic data interpretations.

- **Over-Saturation in Score Distillation.** The problem of over-saturation within score distillation algorithms remains insufficiently explored and unresolved. This issue manifests as a discrepancy in color distribution between the 3D content created through score distillation techniques and the 2D images generated by diffusion inference processes. Although certain methodologies yield results with more appropriate levels of saturation, they impose a significant computational load. Examples include LoRA finetuning [53], adversarial training [5], or supervision in multiple spaces with iterative sampling [62]. Thus, a comprehensive understanding of the over-saturation problem and a fast yet effective score distillation algorithm is needed.

To overcome these challenges, we propose PlacidDreamer, a framework that harmonizes initialization, multi-view generation, and text-conditioned generation with a single multi-view diffusion model, while simultaneously employing a novel score distillation algorithm to achieve balanced saturation. More specifically, to address the first limitation, we newly devised a Latent-Plane module. This module enhances the multi-view diffusion model by enabling fast geometry reconstruction and improving its capabilities in generating multi-view images. The reconstructed geometry is then utilized to initialize 3D Gaussian points, and the improved multi-view images are used to personalize the text-to-image diffusion model with directional prompts. This coordinated approach aligns the generation directions within the pipeline with the outcomes from the multi-view diffusion model, promoting convergence and

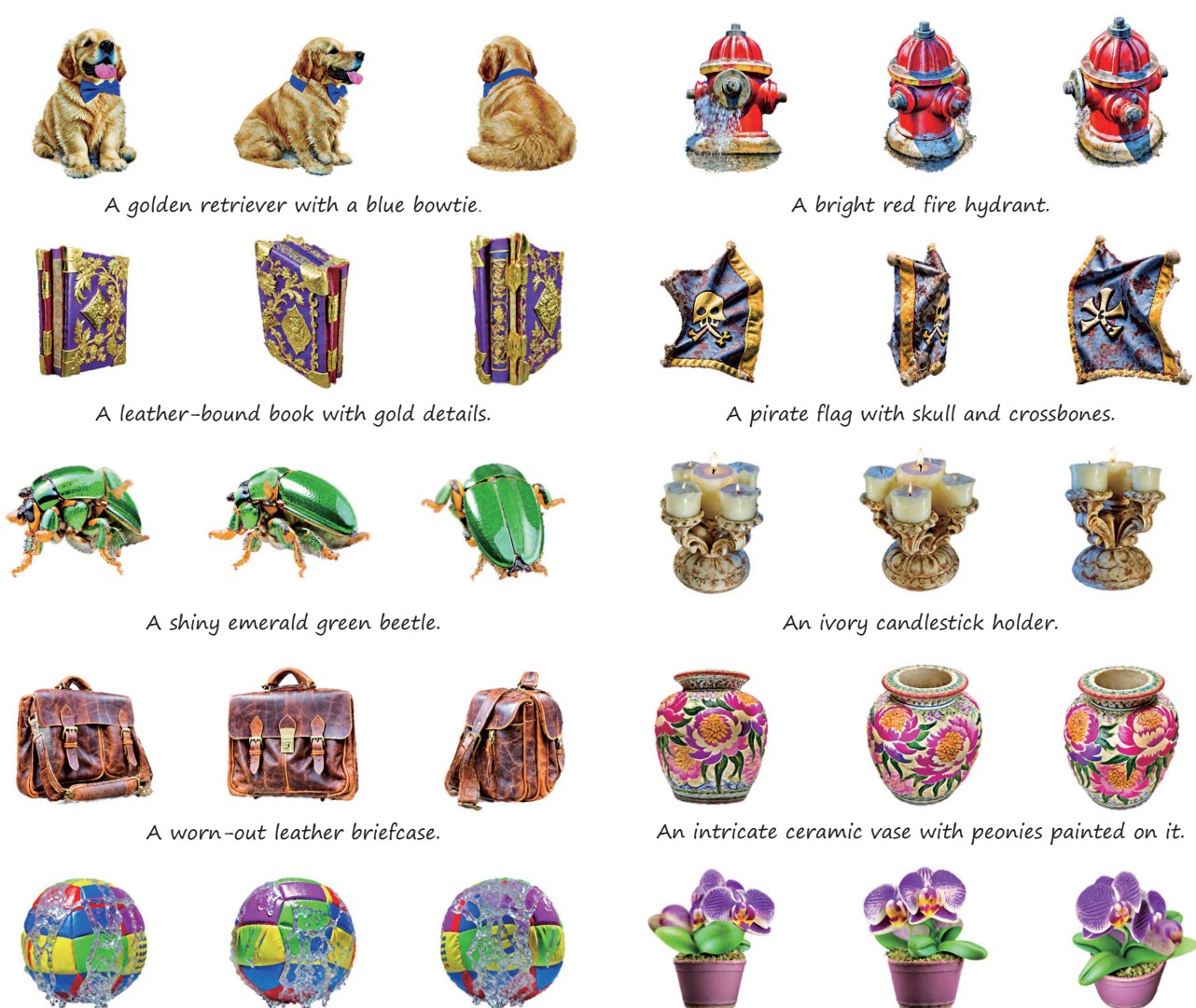

A golden retriever with a blue bowtie.

A bright red fire hydrant.

A leather-bound book with gold details.

A pirate flag with skull and crossbones.

A shiny emerald green beetle.

An ivory candlestick holder.

A worn-out leather briefcase.

An intricate ceramic vase with peonies painted on it.

A wet, vibrant beach ball.

A blooming potted orchid with purple flowers.

**Figure 1: 3D generations of PlacidDreamer. More results are provided in the Supplementary Material.**

significantly enhancing the quality of the generated content. Additionally, the Latent-Plane module is designed to be training-efficient and is adaptable to a variety of multi-view diffusion models, accommodating different viewpoint configurations. To address the second limitation, we delve into the causes of over-saturation. We decompose the score distillation equation into two primary components: classifier guidance and smoothing guidance. Prior algorithms have suffered from an imbalance where classifier guidance overwhelmingly dominates smoothing guidance, leading to prevalent over-saturation. Our analysis reveals that in over 30% of instances, the optimization directions of these two guidances form obtuse angles, resulting in negative optimization along certain directions with a fixed Classifier-Free Guidance (CFG) parameter, making it challenging to maintain control over balance after multiple optimization steps. To rectify this imbalance, we propose to treat score distillation as a multi-objective optimization problem and introduce

a Balanced Score Distillation (BSD) algorithm. This algorithm incorporates a multi-objective optimization solution, Multiple-Gradient Descent Algorithm (MGDA), which dynamically adjusts the optimization directions to converge at Pareto Optimal points, where the generated results exhibit both rich details and balanced saturation.

We evaluate PlacidDreamer both qualitatively and quantitatively to demonstrate its effectiveness. Extensive experiments indicate the superiority of our method over previous methods. Quantitative evaluations conducted on the T3Bench [10] benchmark reveal that PlacidDreamer consistently outperforms baseline methods by a margin of at least ten points in both generation quality and alignment metrics. Furthermore, we conduct ablation studies to respectively evaluate the contribution of each proposed module to the overall quality. To further highlight the BSD algorithm's effectiveness, we replaced the score distillation algorithms in various open-source text-to-3D frameworks with BSD, while keeping other

components constant. Results show that BSD consistently enhances these frameworks' performance.

Our contributions can be summarized as follows:

- We introduce PlacidDreamer, a novel framework designed for high-fidelity text-to-3D generation. PlacidDreamer advances a more harmonious generation process through two novel approaches: the Latent-Plane module, which enhances multi-view diffusion models, and the Balanced Score Distillation algorithm, which optimizes detail richness and saturation control.
- We identify the causes of the over-saturation problem in score distillation algorithms and propose to view score distillation as a multi-objective optimization problem, with the goal of optimizing towards Pareto Optimal points to stabilize the outcomes of generation.
- Extensive experiments, including both quantitative metrics and qualitative assessments, demonstrate that PlacidDreamer significantly outperforms existing state-of-the-art methods.

## 2 RELATED WORKS

**Text-to-3D Generation.** DreamFields [18] initially employs a pre-trained model, CLIP [40], to guide the optimization of NeRF [34]. Trying to leverage the generative nature of the diffusion model, Dreamfusion [37] introduces SDS, a loss associated with a score function derived from distilling 2D diffusion models. Subsequent works significantly improve SDS-based methods, including those using multi-view consistent models [24, 30, 44, 61], enhancing pipeline structures [4, 26, 32, 33, 43, 54, 58], introducing extra generation priors [2, 15, 17, 19, 48, 56, 60], or exploring timestep scheduling [16]. Recently, some fast-forward reconstruction models [13, 28, 30, 31, 50] have emerged for faster 3D generation.

**Score Distillation.** Given the heavy reliance on score distillation methods in these studies, addressing associated issues with SDS is crucial. While many works [11, 21, 48, 51, 52, 59] focus on addressing over-smoothing, fewer tackle over-saturation problems. ProlificDreamer [53] introduces VSD, leveraging LoRA finetuning for 3D distribution modeling, which can also alleviate over-saturation. IT3D [5] handles over-saturation by training a discriminator distinguishing 3D assets from text-to-2D images. Additionally, HiFA [62] presents an iterative score distillation process for a more accurate sampling direction. It's worth noting that these methods introduce a significant computational burden, slowing down overall speed.

## 3 PRELIMINARIES

### 3.1 Diffusion Models

For discrete-time diffusion models [9, 12, 36, 45], given a data distribution $q(\mathbf{x}) = q_0(\mathbf{x}_0)$, we construct a forward process with a series of distributions $q_t(\mathbf{x}_t) = \mathcal{N}(\alpha_t \mathbf{x}_0, (1 - \alpha_t^2)\mathbf{I})$ with decreasing $\{\alpha_t | t \in [0, T], t \in \mathbb{Z}\}$, where $\alpha_0 = 1$ and $\alpha_T \approx 0$. In the generation process, we start from $\mathbf{x}_T \sim \mathcal{N}(\mathbf{0}, \mathbf{I})$ and generate a sample of the previous timestep iteratively with a denoising network $\epsilon_\phi(\mathbf{x}_t, t)$ trained by minimizing the prediction of added noise, which is given by

$$\mathbb{E}_{\mathbf{x}_0 \sim q_0(\mathbf{x}_0), t, \epsilon \sim \mathcal{N}(\mathbf{0}, \mathbf{I})} w(t) \|\epsilon_\phi(\alpha_t \mathbf{x}_0 + \sigma_t \epsilon, t) - \epsilon\|_2^2, \quad (1)$$

where $w(t)$ is to balance losses between different timesteps, $t$ is uniformly selected from 0 to $T$, and $\sigma_t = \sqrt{1 - \alpha_t^2}$.

### 3.2 Score Distillation Sampling (SDS)

Score distillation is an optimization-based generation method that distills knowledge from pre-trained 2D diffusion models to guide other generations, The first score distillation method, SDS [37] is denoted as

$$\nabla_\theta \mathcal{L}_{SDS}(\phi, \mathbf{x} = g(\theta)) \triangleq \mathbb{E}_{t,\epsilon} [\omega(t)(\hat{\epsilon}_\phi(\mathbf{x}_t; y, t) - \epsilon)\frac{\partial \mathbf{x}}{\partial \theta}], \quad (2)$$

where $\mathbf{x}_t = \alpha_t \mathbf{x} + \sigma_t \epsilon$ and other symbols are defined the same as in Equation (1). Intuitively, this loss perturbs $\mathbf{x}$ with a random amount of noise corresponding to the timestep t, and estimates an update direction that follows the score function of the diffusion model to move to a higher density region.

### 3.3 Multiple-Gradient Descent Algorithm

MGDA is a gradient-based algorithm used to solve multi-objective optimization problems. Multi-objective optimization refers to the task of finding a Pareto Optimal solution under multiple optimization criteria, where optimizing one objective does not deteriorate the solution of another objective during the optimization process. In the context of solving multi-objective optimization tasks, MGDA determines a descent direction that is common to all criteria. For instance, when applied to a binary-objective optimization problem optimizing $\mathcal{L}^1(x)$ and $\mathcal{L}^2(x)$, MGDA will identify the direction orthogonal to $\nabla_x \mathcal{L}^1(x) - \nabla_x \mathcal{L}^2(x)$. The algorithm has been demonstrated to converge to a Pareto Optimal point, where there is no available optimization that can improve one objective without worsening another. Further elaboration on MGDA can be found in the work of Désidéri [8].

## 4 METHODS

### 4.1 Pipeline

Given a text prompt, we initially generate a reference image using Stable Diffusion [41] or MVDream [44]. Unlike previous methods that transform text-to-3D generation into single-view reconstruction, we do not aim to precisely reconstruct the reference object. After background removal, the reference image is fed into the multi-view diffusion model to generate multi-view images. To optimize the initialization of 3D Gaussian points [22] and ensure its compatibility with the multi-view images, we introduce Latent-Plane. This module, as detailed in Section 4.2, seamlessly integrates into the latent layers of any multi-view diffusion model, enabling fast reconstruction of a volume density field and enhancement of the generated images within 40 seconds. Moreover, it is training-friendly (only requiring $4 \times$ A6000 GPUs for 18 hours), as it directly leverages 3D spatial knowledge from pre-trained multi-view diffusion model. Subsequently, the volume density field is used to initialize the 3D Gaussian points on the object surface, while the generated multi-view images are used to fine-tune Stable Diffusion with LoRA [14] technique. During this fine-tuning process, directional prompts such as "left view" are incorporated into the original text, enhancing the diffusion model's awareness of 3D space. Finally, we supervise the splatted images using the BSD guidance from the fine-tuned

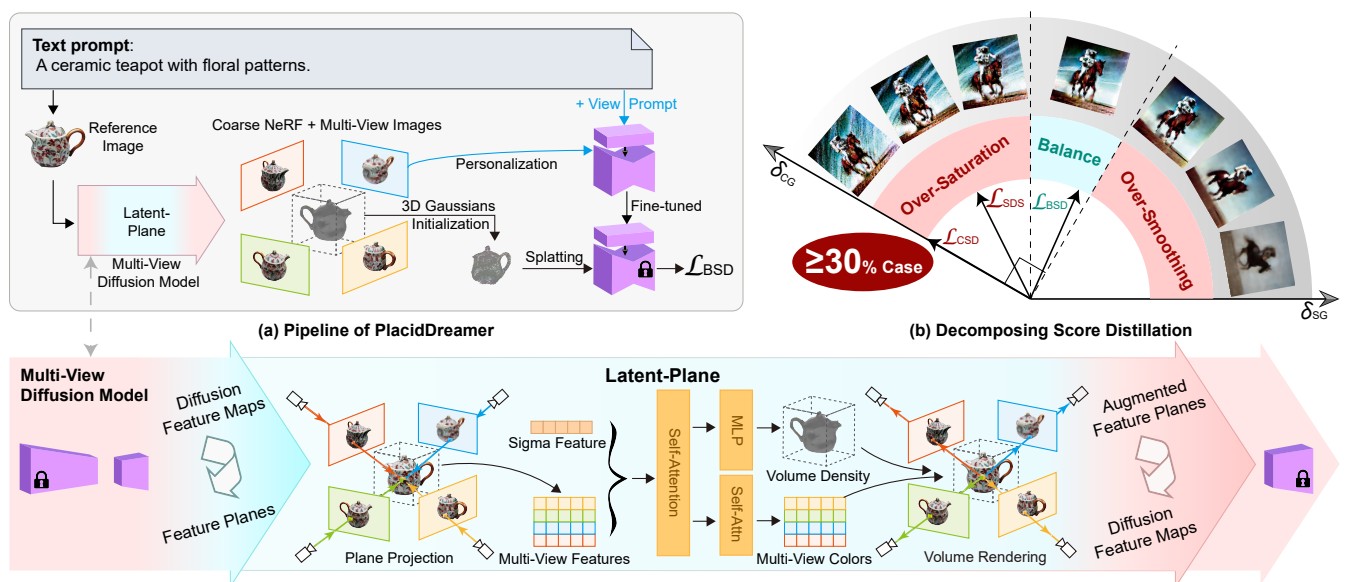

Figure 2: (a) The pipeline of PlacidDreamer. (b) Score distillation can be decomposed into two directions: classifier guidance $\delta_{CG}$ and smoothing guidance $\delta_{SG}$. CSD [59] only utilizes classifier guidance. In more than 30% of cases, the angle between these two guidance vectors is obtuse. In such scenarios, using a fixed CFG parameter in SDS may result in negative optimization in the $\delta_{SG}$ direction, leading to over-saturation. However, BSD algorithm ensures that each optimization step is non-negative in both directions. (c) The integration of the Latent-Plane module with multi-view diffusion models.

diffusion model. As elaborated in Section 4.3, we decompose score distillation and reveal a conflict between optimization directs, leading to color discrepancies. We suggest treating score distillation as a multi-objective optimization problem and we introduce the BSD algorithm, achieving rich details and reasonable colors.

## 4.2 Latent Plane

The proposed method, Latent-Plane, serves as a plug-in module compatible with various multi-view diffusion models. These diffusion models typically take an input image $I_{\pi_0}$ of an object from a specific viewpoint $\pi_0$ and generate corresponding images $I_{\pi_i}$ from other viewpoints $\pi_i, i = 1, 2, \ldots, N$. Inspired by ConsisNet [57], which reinforces feature patches by utilizing coordinate relationships derived from back-projection, we hypothesize that pretrained features within multi-view diffusion models possess sufficient knowledge of the 3D space to conduct direct reconstruction. Leveraging this idea, we treat latent feature maps akin to the feature plane in the Tri-Plane [3] model, allowing us to reconstruct a volume density field. Furthermore, the reconstruction results can be utilized to enhance the latent feature map through volume rendering, thereby improving the accuracy of multi-view predictions.

*4.2.1 Multi-View Feature Gathering.* Most multi-view diffusion models currently utilize the Unet [42] architecture for their denoisers. To illustrate our strategy of selecting the appropriate latent layer for inserting the Latent-Plane module, we consider the Unet architecture as an exemplar. Within the Unet architecture, assuming there are $L$ decoder blocks generating $L$ feature maps denoted as $\mathbf{F}_j \in \mathbb{R}^{H \times W \times D}$ with $j$ indexing these maps and $D$ representing the feature dimension. We choose the feature map with the highest

resolution and the smallest index, denoted as $\mathbf{F}_k$, as it encompasses the deepest and most intricate features extracted by the model, containing comprehensive geometric information.

At each timestep $t$, during the reverse diffusion process, we obtain $N$ feature maps $\mathbf{F}_k^{(i)}, i = 1, 2, \ldots, N$ from $N$ viewpoints. For every point $\mathbf{x}$ in the 3D space, we project it onto each feature map $\mathbf{F}_k^{(i)}$ and derive its corresponding feature $\mathbf{f}_k^{(i)}(\mathbf{x})$ through tri-linear interpolation, denote as,

$$\mathbf{f}^{(i)}(\mathbf{x}) = \text{Interp\_2D}(\text{Proj}(\mathbf{x}, \mathbf{F}_k^{(i)}), \mathbf{F}_k^{(i)}), \quad (3)$$

where Interp\_2D represents the 2D interpolation function, and Proj denotes the function for projecting 3D spatial points onto 2D planes. We then apply an additional linear layer to extract low-dimensional features from the extracted features. To better represent the features of spatial point $\mathbf{x}$, we concatenate each feature with its respective camera embeddings $\mathbf{e}_{cam}^{(i)}$ and coordinate embeddings $\mathbf{e}_{pos}(\mathbf{x})$.

$$\mathbf{e}^{(i)}(\mathbf{x}) = \text{Concat}[\text{Linear}(\mathbf{f}^{(i)}(\mathbf{x})), \mathbf{e}_{cam}^{(i)}, \mathbf{e}_{pos}(\mathbf{x})]. \quad (4)$$

*4.2.2 Plane-Based NeRF.* So far, $\mathbf{x}$ has $N$ features gathered from $N$ viewpoints, serving as $N$ tokens, which will be mutually enhanced through the Attention layers. To obtain the volume density $\sigma(\mathbf{x})$ for the spatial point, before feeding into the attention layer, we additionally add a token $\tau_\sigma$ corresponding to obtaining sigma. $\tau_\sigma$ is obtained by concatenating a trainable embedding $\mathbf{e}_\sigma$ with embeddings of timestep $t$, followed by a linear layer.

$$\tau_\sigma = \text{Linear}(\text{Concat}[\mathbf{e}_\sigma, \mathbf{e}_t]), \quad (5)$$

$$\tau(\mathbf{x}) = \text{Concat}[\tau_\sigma, \mathbf{e}^{(1)}(\mathbf{x}), \mathbf{e}^{(2)}(\mathbf{x}), \ldots, \mathbf{e}^{(N)}(\mathbf{x})], \quad (6)$$

$$\tau'(\mathbf{x}) = \text{MultiHeadSelfAttn}(\tau(\mathbf{x})), \tag{7}$$

where MultiHeadSelfAttn denotes the Multi-Head Self Attention layers. After the multi-view augmentation, there are a total of N+1 tokens in $\tau'(\mathbf{x})$. For the sigma token $\tau'_\sigma(\mathbf{x})$, we use an MLP to obtain its volume density value $\sigma(\mathbf{x})$. For the multi-view enhanced feature tokens $\tau'(\mathbf{x})$, we conduct volume rendering to gather features enhanced on the object surface.

$$\sigma(\mathbf{x}) = \text{MLP}(\tau'_\sigma(\mathbf{x})). \tag{8}$$

$$\mathbf{F}'^{(i)}_k = \text{Volume\_Rendering}(\sigma(\mathbf{x}), \tau'^{(i)}(\mathbf{x})) \tag{9}$$

We directly add the feature map to the original feature map. The enhanced feature map continues to function in the decoder of the diffusion model, ultimately producing the model's predicted noise $\epsilon$, which is then integrated into diffusion training and inference.

*4.2.3 Training.* The training comprises two stages. In the first stage, we independently train the volume density generation module, enabling the selection of arbitrary camera viewpoints, rather than being constrained to predefined fixed camera viewpoints. Subsequently, we can calculate the depth of each spatial point relative to the selected camera. Following volume rendering, we supervise the generated occupancy map $M^{(i)}$ and depth maps $D^{(i)}$, which are denoted as

$$\mathcal{L}_\sigma = \mathcal{L}_{\text{BCE}}(M^{(i)}, M^{(i)}_{\text{gt}}) + \lambda(1 - \rho(D^{(i)}, D^{(i)}_{\text{gt}})), \tag{10}$$

where, $\mathcal{L}_{\text{BCE}}$ denotes the Binary Cross Entropy loss, $\rho$ represents the Pearson correlation coefficient, and $\lambda$ controls the balance between the two losses. For training the feature map augmentation, we use the standard diffusion loss described in Equation (1).

## 4.3 Balanced Score Distillation

In this chapter, our exploration of score distillation is conducted through 2D generation experiments, where we directly optimize the pixel values of a 2D image. The use of 2D experiments allows the exclusion of various factors unrelated to score distillation algorithms, such as camera viewpoints and 3D representation selection, thereby highlighting the effectiveness of score distillation methods. Furthermore, the outcomes of 2D experiments can effectively represent the effects observed in 3D experiments [11, 53]. Our focus on the experimental results is specifically directed toward the richness of details and color saturation at two distinct time points: the optimal state of image quality (referred to as the convergence state) and the period of sustained training after reaching the convergence state (referred to as the over-trained state). In 3D generation, the lack of an early-stop mechanism frequently results in the widespread occurrence of over-training.

*4.3.1 Decomposing Score Distillation.* We adopt a modeling approach different from SDS [37] to obtain our decomposition. Score distillation is an optimization-based generation method aimed at distilling knowledge from pre-trained 2D diffusion models to guide other generations. In order to utilize 2D diffusion models for guidance, it is necessary to establish a connection between the 2D image distribution $p_0$ modeled by diffusion models and the 3D representation distribution $q$. We make the assumption that the probability density distribution $q(\theta|y)$ for the parameters $\theta$ of the 3D representation, conditioned on the text prompt $y$, is proportional to the product of the conditional probability densities of its rendered images $\mathbf{x}_0^\pi$ under various viewpoint $\pi$s, denoted as,

$$q(\theta|y) \propto \prod_\pi p_0(\mathbf{x}_0^\pi(\theta)|y). \tag{11}$$

We adopt the average negative logarithm of the probability,

$$\mathcal{L}_{\text{SD}} = -\frac{1}{N} \log q(\theta|y), \tag{12}$$

as the loss for neural network training, where $N$ is the number of $\pi$s. Consequently, as the loss decreases, we obtain a sample as described in the text prompt. We can get the gradient on $\theta$ as,

$$\nabla_\theta \mathcal{L}_{\text{SD}} = -\mathbb{E}_\pi [\nabla_\theta \log p_0(\mathbf{x}_0^\pi(\theta)|y)]. \tag{13}$$

However, $p_0$ represents the unknown distribution of general 2D images, and high-density regions of $p_0$ are sparsely populated [46]. Given that current diffusion models are designed to model the score function of the distribution $p_t$ of noisy images with noise of level $t$, we instead leverage it to generate an optimized gradient for the noisy images $\mathbf{x}_t$ that is obtained by adding noise of level $t$ to $\mathbf{x}_0^\pi$. Subsequently, this gradient can be back-propagated to the image $\mathbf{x}_0^\pi$ and further transmitted to the 3D parameters $\theta$, denoted as,

$$\nabla_\theta \mathcal{L}_{\text{SD}} = -\mathbb{E}_{\pi,t} [\nabla_{\mathbf{x}_t} \log p_t(\mathbf{x}_t(\theta)|y) \frac{\partial \mathbf{x}_t}{\partial \mathbf{x}_0^\pi} \frac{\partial \mathbf{x}_0^\pi}{\partial \theta}]. \tag{14}$$

Please note that we observe a distinction between our formula and Equation (2) from SDS [37], particularly in the absence of the final term $-\epsilon$ in our formulation. The first term $\nabla_{\mathbf{x}_t} \log p_0(\mathbf{x}_t(\theta)|y)$ is a 2D score function modeled by diffusion models. In 2D diffusion generation, the score function of conditional distribution can be decomposed into the classifier guidance and the score function of an unconditional distribution:

$$\nabla_{\mathbf{x}_t} \log p_t(\mathbf{x}_t|y) = \nabla_{\mathbf{x}_t} \log p_t(y|\mathbf{x}_t) + \nabla_{\mathbf{x}_t} \log p_t(\mathbf{x}_t). \tag{15}$$

So, we naturally decompose the score distillation process into two functional terms.

$$\delta_{\text{CG}} = -\nabla_{\mathbf{x}_t} \log p_t(y|\mathbf{x}_t) = (\epsilon(\mathbf{x}_t, t, y) - \epsilon(\mathbf{x}_t, t, \emptyset))/\sigma_t \tag{16}$$

$$\delta_{\text{SG}} = -\nabla_{\mathbf{x}_t} \log p_t(\mathbf{x}_t) = \epsilon(\mathbf{x}_t, t, \emptyset)/\sigma_t \tag{17}$$

$$-\nabla_{\mathbf{x}_t} \log p_t(\mathbf{x}_t|y) = u \cdot \delta_{\text{CG}} + v \cdot \delta_{\text{SG}}, \tag{18}$$

where $u$ and $v$ control the ratio between the two terms. The first term is the classifier guidance term ($\delta_{\text{CG}}$). When only utilizing classifier guidance, referred to as CSD [59], images exhibit fine details but suffer from artifacts and over-saturation, as shown in the upper right corner of Figure 2. Therefore, the introduction of the second term aims to alleviate the artifacts caused by $\delta_{\text{CG}}$ and guide $\mathbf{x}_0$ towards the higher density region in the distribution of 2D images. As depicted in Figure 2, starting from $\mathcal{L}_{\text{CSD}}$, as $u/v$ decreases, the optimization directions move towards the "balance" area, the generation results become smoother. Hence, we refer to it as smoothing guidance ($\delta_{\text{SG}}$). However, when $\delta_{\text{CG}}$ starts to dominate the proportion, the image tends to become overly smoothed.

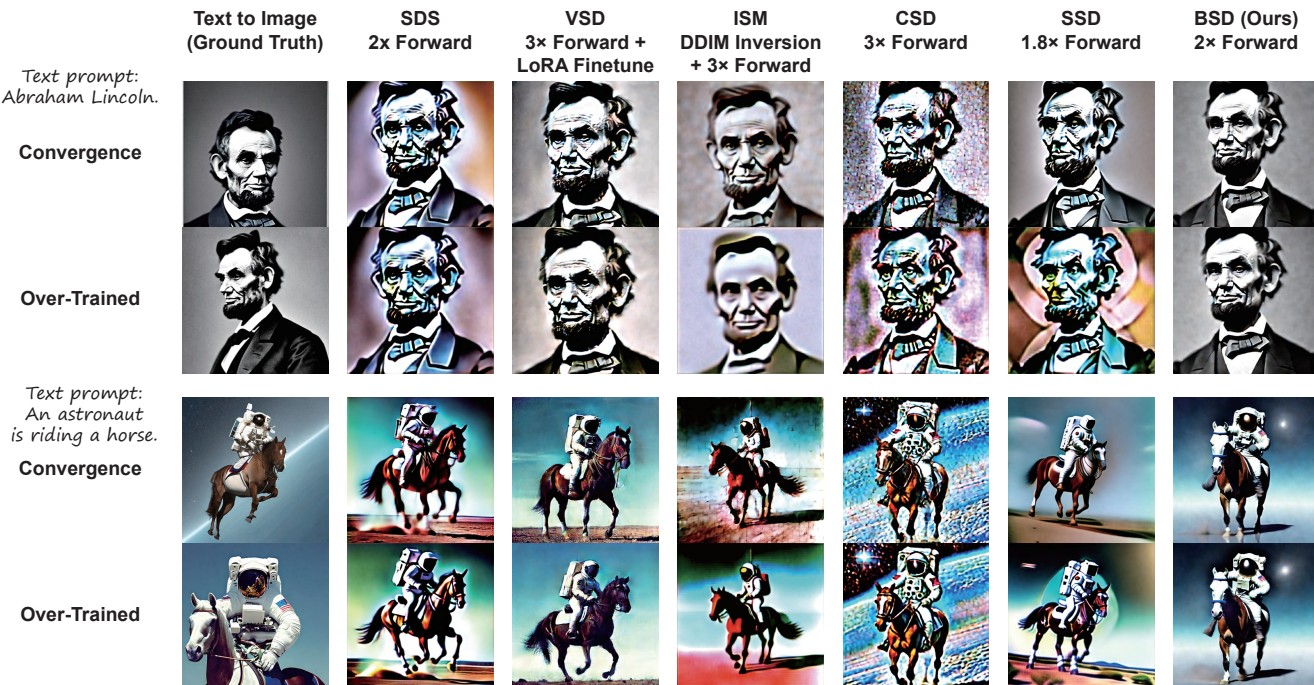

| | Text to Image (Ground Truth) | SDS 2× Forward | VSD 3× Forward + LoRA Finetune | ISM DDIM Inversion + 3× Forward | CSD 3× Forward | SSD 1.8× Forward | BSD (Ours) 2× Forward |

**Figure 3: 2D generation results of score distillation algorithms, annotated with computational costs. "Forward" represents undergoing one forward process of the diffusion model. The results of our BSD closely resemble the text-to-image ground truth. BSD converges at the Pareto Optimal points, ensuring that its results maintain balanced saturation during over-training.**

*4.3.2 Balanced Score Distillation.* Based on our analysis, the parameters of CFG ($u/v$) in score distillation algorithms play a critical role in regulating the balance between the two guidance terms during each optimization step. However, we have observed that their effectiveness becomes highly unstable after numerous optimization steps, particularly highlighted by the failure of most score distillation methods to preserve image properties during the over-training phase. We conduct experiments of previous score distillation algorithms including SDS [37], VSD [53], ISM [25], CSD [59], SSD [48]. We find that most of them exhibit over-saturation during over-trained steps, as depicted in Figure 3.

To address this issue, we investigated the distribution characteristics of $\delta_{\text{CG}}$ and $\delta_{\text{SG}}$. We observed that over 30% of the steps in the iterative optimization process exhibit a negative dot product between $\delta_{\text{CG}}$ and $\delta_{\text{SG}}$, indicating obtuse angles between their optimization directions in the high-dimensional space, as shown in Figure 2. In such instances, simply combining the two directions based on a fixed ratio may lead to the final optimization direction projecting negatively onto one of the term's optimization directions. Consequently, the control of balance becomes ineffective due to the presence of negative optimization, ultimately resulting in over-saturation or over-smoothing.

Therefore, we propose to consider score distillation as a multi-objective optimization task, where the optimization objectives are $\mathcal{L}^1 = -\lambda \log(p_t(y|\mathbf{x}_t))$ and $\mathcal{L}^2 = -\log(p_t(\mathbf{x}_t))$, with $\lambda$ representing the hyper-parameter that controls the ratio of two guidance terms. The solution of multi-objective optimization tasks will find the Pareto optimal point, where no action can improve one objective without deteriorating another. If generation reaches the

Pareto Optimal points, it will stabilize at this optimal point, and the generated content will exhibit rich details while maintaining balanced saturation. In this scenario, the hyper-parameter $\lambda$ genuinely controls the final inclination towards both optimization directions, rather than attempting to control the balance at each step.

We propose the Balanced Score Distillation, which employs MGDA [8]. Suppose the optimization combination of $\nabla_{\mathbf{x}_t}\mathcal{L}^1$ and $\nabla_{\mathbf{x}_t}\mathcal{L}^2$ is

$$\nabla_{\mathbf{x}_t} \log \tilde{p}_t(\mathbf{x}_t|c) = \alpha \nabla_{\mathbf{x}_t}\mathcal{L}^1 + (1-\alpha)\nabla_{\mathbf{x}_t}\mathcal{L}^2. \quad (19)$$

Following Désidéri [8], $\alpha$ is the solution of

$$\min_{\alpha, 1-\alpha \geq 0} \left\{ \left\| \alpha \nabla_{\mathbf{x}_t}\mathcal{L}^1 + (1-\alpha)\nabla_{\mathbf{x}_t}\mathcal{L}^2 \right\|_2^2 \right\}. \quad (20)$$

For binary situation like in Equation (20), we have closed form solution for $\alpha$, which is $\alpha = \min(\max(0, \hat{\alpha}), 1)$, where

$$\hat{\alpha} = \frac{(\nabla_{\mathbf{x}_t}\mathcal{L}^2 - \nabla_{\mathbf{x}_t}\mathcal{L}^1)^T \nabla_{\mathbf{x}_t}\mathcal{L}^2}{\|\nabla_{\mathbf{x}_t}\mathcal{L}^2 - \nabla_{\mathbf{x}_t}\mathcal{L}^1\|_2^2}. \quad (21)$$

Thus, the final formula for BSD is:

$$\nabla_\theta \mathcal{L}_{\text{SD}} = \mathbb{E}_{\pi,t}[\omega(t)(\alpha\lambda\delta_{\text{CG}} + (1-\alpha)\delta_{\text{SG}})\frac{\partial \mathbf{x}_0^\pi}{\partial \theta}], \quad (22)$$

$$\alpha = \min\left[\max\left[0, \frac{(\delta_{\text{SG}} - \lambda\delta_{\text{CG}})^T \delta_{\text{SG}}}{\|\delta_{\text{SG}} - \lambda\delta_{\text{CG}}\|_2^2}\right], 1\right]. \quad (23)$$

As illustrated in Figure 3, images generated by the BSD algorithm approximate the saturation of 2D images, showcasing smooth color transitions and detailed contours. In states of over-training, BSD maintains a harmonized saturation level, unlike other methods that suffer from loss of details due to over-saturation. A comparable

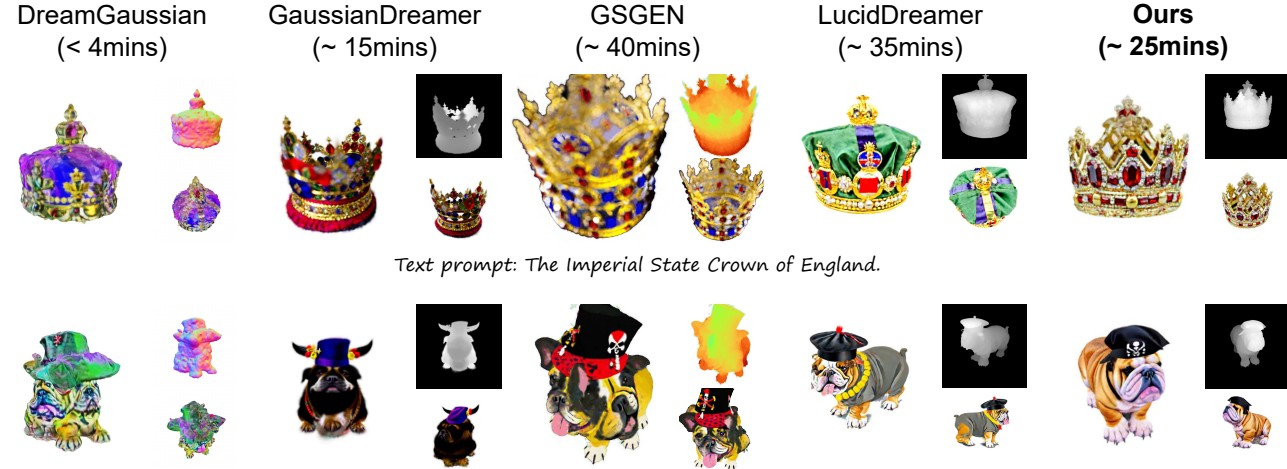

DreamGaussian (< 4mins) | GaussianDreamer (~ 15mins) | GSGEN (~ 40mins) | LucidDreamer (~ 35mins) | **Ours (~ 25mins)**

Text prompt: The Imperial State Crown of England.

Text prompt: A bull dog wearing a black pirate hat.

**Figure 4: Qualitative comparison with baseline methods. More comparisons with recent methods utilizing NeRF [34] as 3D representations are provided in the Supplementary Material.**

method to BSD is VSD [53], which exhibits fewer instances of over-saturation and finer details compared to other baseline methods. However, as annotated in the figure, VSD requires significantly longer computational time and higher GPU utilization than BSD.

*4.3.3 Relationship with Previous Methods.* We provide an analysis of the relationship between our decomposition and score distillation algorithm with previous works in the Supplementary Material.

## 5 EXPERIMENTS

**Table 1: Results of quantitative comparison with baselines and ablation studies. The values in parentheses represent the direct scores given by the ImageReward [55] model.**

|  | Quality | Alignment | Average |
|---|---|---|---|
| DreamFusion | 24.9 (-1.255) | 24.0 | 24.4 |
| Magic3D | 38.7 (-0.565) | 35.3 | 37.0 |
| LatentNeRF | 34.2 (-0.790) | 32.0 | 33.1 |
| Fantasia3D | 29.2 (-1.040) | 23.5 | 26.4 |
| SJC | 26.3 (-1.185) | 23.0 | 24.7 |
| ProlificDreamer | 51.1 (+0.055) | 47.8 | 49.4 |
| PlacidDreamer (Ours) | **62.4 (+0.620)** | **59.8** | **61.1** |
| w/o Latent-Plane | 56.5 (+0.325) | 54.6 | 55.6 |
| w/o Initialization | 57.3 (+0.365) | 56.1 | 56.7 |
| w/o Personalization | 61.8 (+0.590) | 59.5 | 60.7 |
| w/o BSD | 60.2 (+0.510) | 57.4 | 58.8 |

### 5.1 Implementation Details

The Latent Plane model has $N = 8$ viewpoints. Given the resolution limitation of feature map $\mathbf{F}^{(2)}$ to $32 \times 32$, it becomes imperative to regard the pixel space as continuous, ensuring accurate calculation of boundary correspondence. Even a minor difference of 0.5 pixels along the boundary could lead to notable errors. For all the MLP described in Section 4, we use a single linear layer for computing efficiency. We train the Latent-Plane module with Zero-1-to-3 [29]

for 18 hours on $4 \times$ Nvidia RTX A6000 GPUs with a batch size of 32 and loss scale $\lambda = 0.05$ with Objaverse Dataset [7] rendered by SyncDreamer [30], filtered by RichDreamer [39] and LGM [49]. For more details, please refer to the Supplementary Materials.

### 5.2 Comparing with Baselines.

**Quantitative comparison.** T3Bench [10] serves as a benchmark for text-to-3D generation, standardizing camera poses for rendering images and utilizing pre-trained models [1, 23, 40, 55] to ensure a fair evaluation. The primary 3D representation employed in this benchmark is a textured mesh, which aids in the convergence of generated 3D samples, forming seamless surfaces. However, extracting meshes from 3D Gaussian representations poses challenges. Consequently, we employ 3D Gaussian splatting to generate RGB images directly. This choice challenges our generation's quality by not eliminating noise in direct RGB image generation.

As PlacidDreamer focuses on the generation of individual objects, our evaluations are confined to the single-object part of T3Bench. Considering DreamFusion [37], Magic3D [26], Fantasia3D [4], ProlificDreamer [53], Latent-NeRF [33], and SJC [51] as baseline methods, all of which were tested by T3Bench authors, He et al. [10]. The outcomes reveal that PlacidDreamer significantly outperforms the baseline methods in both generative quality and accuracy corresponding to the provided text. Our average quality score surpasses the next-highest ProlificDreamer by 11 points. Since this score is a linear transformation of the ImageReward [55] score, we annotate the original ImageReward scores, providing a clearer perspective on the superiority of our generative quality.

**Qualitative comparison.** We also conduct qualitative comparisons on several baseline methods which also builds on 3D Gaussian Splatting [22], including DreamGaussian [50], GaussianDreamer [58], GSGEN [6], and LucidDreamer [25]. In Figure 4, we present the results of different methods generating the same prompts. DreamGaussian can generate a textured mesh in several minutes, thus, its generation quality is lower compared with other methods. GaussianDreamer and GSGEN leverages end-to-end 3D generation model

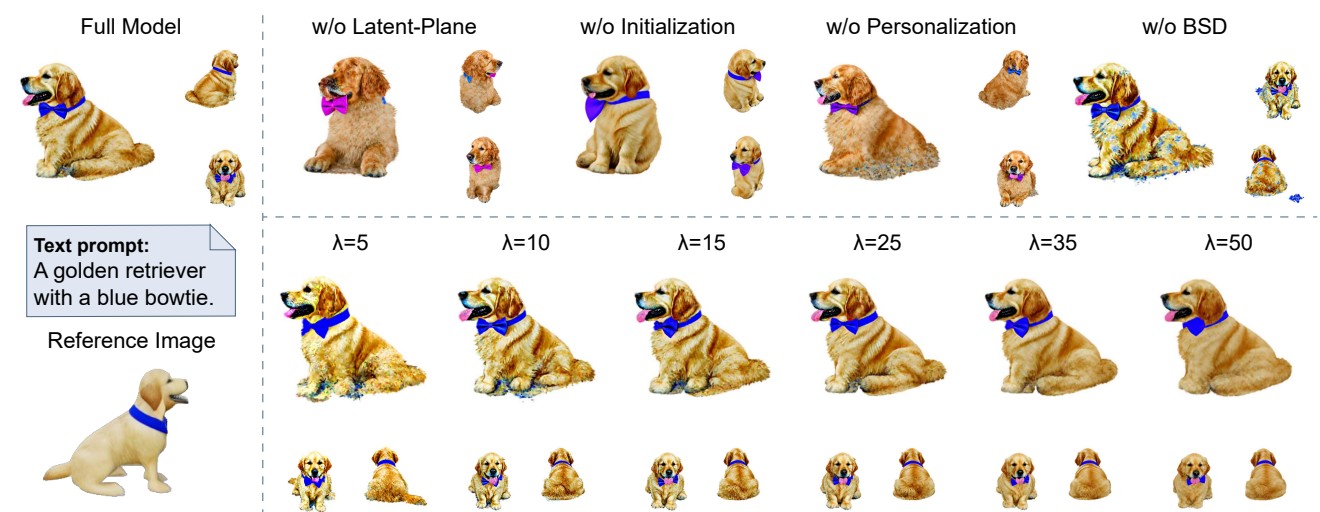

**Figure 5: Results of ablation studies. In the first line, we evaluate PlacidDreamer by removing each component individually. In the second line, we investigate the impact of different $\lambda$ values, validating that BSD enables stable control of the balance between color saturation and detail level.**

[20, 35] for initialization with multi-view consistency. However, their final results still have multi-face problems, because the 3D model and Stable Diffusion [41] generate towards different directions. LucidDreamer's generations feature bright colors according to the ISM algorithm and exhibit good geometric structure. However, with Latent-Plane enhancing texture and the BSD algorithm, PlacidDreamer achieves superior textures and richer details.

## 5.3 Ablation Studies.

We perform ablation studies, as illustrated in Figure 5, and carry out quantitative evaluations using T3Bench, as detailed in Table 1. **Latent-Plane.** The "Latent-Plane" module in the pipeline serves the functions of initializing 3D Gaussian points and personalizing the diffusion model. We investigate the effects of removing both functionalities (w/o Latent-Plane), removing only initialization (w/o Initialization), and removing only fine-tuning (w/o Personalization) on the experimental results. Firstly, we remove the Latent-Plane module (w/o Latent-Plane), utilizing the Point-e [35] model as the initialization model for 3D Gaussian points, following Lucid-Dreamer. Point-e struggles to comprehend the complex prompts and fails to generate a complete shape for the dog. Additionally, due to the color deviation introduced by Stable Diffusion during generation, the color of the dog's tie is inaccurate. Subsequently, we only remove the initialization function of the Latent-Plane, retaining the fine-tuning part (w/o Initialization). It can be observed that the Janus problem in the dog's head is resolved. Through extensive experimentation, we find that under the probabilistic supervision of score distillation, 3D Gaussian points do not undergo significant changes in position, making them highly sensitive to initialization. Next, we solely remove the fine-tuning part, keeping the initialization part intact (w/o Personalization). This results in a more complete structure for the dog's bodies, but issues arise with the multi-face problem and tie color.
**BSD.** We replace BSD with SDS [37] and observe severe over-saturation and some blue artifacts. Compared with BSD, SDS makes

3D Gaussian points harder to converge. To further investigate BSD, we conduct ablation experiments on $\lambda$ to validate its effectiveness in controlling the ratio of $\delta_{CG}$ and $\delta_{SG}$. It can be observed that when $\lambda = 5$, the dog's color is over-saturated. As $\lambda$ increases to the range of 15-25, over-saturation alleviates without significant loss of details. Only when $\lambda$ reaches the value of 35 does the loss of details become apparent. However, at this point, the dog's features are still recognizable. It is worth noting that in the SDS algorithm, CFG can only be set to a large value to get recognizable outcomes.

These results prove that each proposed component significantly contributes to the overall effectiveness of the framework.

## 5.4 Experiments on BSD

BSD is a general score distillation method independent of the choice of NeRF, camera perspectives, diffusion models, etc. To test its distillation ability, we integrate BSD into various 3D generation pipelines, ensuring a fair evaluation of score distillation capabilities. This included exclusively substituting alternative score distillation algorithms, without introducing additional modifications to the pipeline. Our results demonstrate that BSD enhances the output of each pipeline. Further details and results are provided in the Supplementary Material.

## 6 CONCLUSION

We have focused on resolving conflicts in current text-to-3D approaches, including conflicts within a single model's guidance and conflicts arising from guidance provided by different models. To address these conflicts, we introduced a novel framework Placid-Dreamer with the newly designed Latent-Plane module and the Balanced Score Distillation algorithm to achieve mutual optimization. As a result, we have achieved a more harmonious and balanced text-to-3D generation process, leading to high-fidelity 3D outcomes. We hope our work will inspire further investigations into more harmonious methodologies for 3D generation.

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
