# OpenReview forum: "PlacidDreamer: Advancing Harmony in Text-to-3D Generation"
_acmmm.org/ACMMM/2024/Conference — MM2024 Poster_

### Official Review · Reviewer_TNJh · 2024-05-03

**Rating:** 5
**Confidence:** 3

**Summary:**

The paper introduces PlacidDreamer,  a framework that harmonizes initialization, multi-view generation, and text-conditioned generation with a single multi-view diffusionmodel, while simultaneously employing a novel score distillation algorithm to achieve balanced saturation. To unify the generation direction, the authors introduce the Latent-Plane module, a plug-in extension that enables multi-view diffusion models to provide fast geometry reconstruction for initialization and enhanced multi-view images to personalize the text-to-image diffusion model. To address the over-saturation problem, they propose to view score distillation as a multi-objective optimization problem and introduce the Balanced Score Distillation algorithm, which offers a Pareto Optimal solution that achieves both rich details and balanced saturation.

**Strengths:**

PlacidDreamer advances a more harmonious generation process through two novel approaches: the Latent-Plane module, which enhances multi-view diffusion models, and the Balanced Score Distillation algorithm, which optimizes detail richness and saturation control.

Extensive experiments indicate the superiority of our method over previous methods.

**Limitations:**

This statement needs support: "In more than 30% of cases, the angle between these two
guidance vectors are obtuse.

Figure 3 lacks a detailed explanation, such as the meaning of "1.8× Forward" in it.

Line 797-800: "The outcomes reveal that PlacidDreamer significantly outperforms the baseline methods in both generative quality and accuracy corresponding to the provided text. Our average quality score surpasses the next-highest ProlificDreamer by 11 points." What metrics are referred to here? Where are the experimental results? Specific explanations are lacking.

**Suitability:**

2

---

### Official Review · Reviewer_g6PL · 2024-05-24

**Rating:** 5
**Confidence:** 2

**Summary:**

This paper introduces PlacidDreamer, a novel framework designed to address limitations in current text-to-3D generation methods. The primary contributions include the Latent-Plane module for harmonizing multi-view and text-conditioned generation and the Balanced Score Distillation (BSD) algorithm to manage over-saturation in generated images. By integrating these innovations, PlacidDreamer aims to unify the generation direction and achieve balanced saturation, enhancing the overall quality and consistency of 3D assets generated from textual descriptions. Extensive experiments validate the framework’s effectiveness, demonstrating superior performance over existing methods.

**Strengths:**

Innovative Framework: PlacidDreamer introduces a comprehensive approach to harmonizing initialization, multi-view generation, and text-conditioned generation. The integration of the Latent-Plane module and BSD algorithm addresses key challenges in the field, such as conflicting optimization directions and over-saturation in score distillation.

Balanced Score Distillation Algorithm: The BSD algorithm treats score distillation as a multi-objective optimization problem, ensuring generated 3D assets have rich details and balanced saturation, which is a significant improvement over previous methods.

Sufficient Evaluation: The paper provides extensive qualitative and quantitative evaluations, including comparisons with baseline methods and ablation studies, demonstrating the superior performance and effectiveness of PlacidDreamer.

**Limitations:**

The complexity of Implementation: The framework's integration of multiple modules and algorithms may introduce complexity, making it challenging to implement and reproduce the results without extensive computational resources.

Dependence on Pre-Trained Models: PlacidDreamer relies heavily on pre-trained 2D and 3D models, which may limit its adaptability and performance in scenarios where such models are not available or not well-suited.

Generalization Across Different Domains: The framework's effectiveness in generating diverse 3D assets across various domains is not thoroughly explored, raising questions about its generalizability and versatility in different application scenarios.

**Suitability:**

2

---

### Official Review · Reviewer_TEH4 · 2024-05-24

**Rating:** 4
**Confidence:** 3

**Summary:**

The authors claim that there are two primary issues in text-to-3D generation: 1) conflicts in generation direction when different diffusion models work in the same framework. 2) over-saturation problem. To overcome the conflicts in generation direction, the authors propose a Latent-Plane module to align the initiation and score distillation with the multi-view diffusion model. To alleviate the over-saturation problem, the author proposes a Balanced Score Distillation algorithm, providing a Pareto Optimal solution.

**Strengths:**

- This paper is well-written.
- The idea of introducing MGDA to adaptively balance classifier guidance and smoothing guidance is interesting.
- The proposed Latent Plane for lifting the multi-view diffusion model to 3D coarse reconstruction may inspire future research.

**Limitations:**

- Several studies have demonstrated the benefits of incorporating a negative prompt term in SDS, like NFSD[1]. I wondering if BCD can handle a ternary-objective optimization problem (classifier, smooth, negative) and how.
- The coarse reconstruction results from Zero-1-to-3 w. Latent-Plane modules are encouraged to be included for a better understanding of the benefits of Latent-Plane.
- Fig. 3 illustrates the efficacy of BSD in addressing over-saturation problems when over-trained. Nevertheless, BSD performs comparably to VSD, indicating the need for a metric to quantify over-saturation for improved comparisons.
- More qualitative comparisons are needed.
- I appreciate the quantitative comparison on T3Bench. However, many methods (e.g., LucidDreamer, MVDream, etc.) are missing in Table 1.

[1] Noise-Free Score Distillation.

Typos:
- Eq. (18): redundant parentheses;

**Suitability:**

3

---

### Official Review · Reviewer_WJz2 · 2024-05-25

**Rating:** 2
**Confidence:** 4

**Summary:**

The paper presents a framework for text-to-3D generation. The key idea is to integrate the Tri-Plane model into the prior multiview diffusion model. The paper also designs a balanced score distillation sampling strategy by leveraging multi-objective optimization theories. Experiments are conducted on T3Bench to validate the effectiveness of the proposal.

**Strengths:**

S1. Well-written and mostly clear.

S2. The proposed balanced score distillation is interesting and new.

**Limitations:**

L1: Integrating volume density field into a multi-view diffusion model is not new. It has been explored by SyncDreamer. The author should discuss the differences between them.

L2. The comparison in Table 1 is not fair. Baseline methods convert the optimized NeRF or DMTet to a textured mesh and then render RGB images from the textured mesh to evaluate generation quality. However, the paper directly uses the rendered image from 3D Gaussian Splatting (3DGS) for evaluation. Note that extracting textured mesh from the original implicit representation (NeRF, DMTet, 3DGS) always has a huge quality degradation. Thus, it is not fair to compare the proposed PlacidDreamer and baseline methods in Table 1. This may be the reason why the average score of PlacidDreamer is significantly higher than baseline methods in Table 1.

L3. The ablation study in Table 1 is also confusing. It seems that the baseline of PlacidDreamer is utilizing Zero-1-to-3 as a prior model and using SDS to optimize the underlying 3DGS. Building upon this, the paper introduces Latent-Plane, Initialization, Personalization, and BSD. Correct me if I am wrong.

L4. The paper trains the Latent-Plane module with Zero-1-to-3 to obtain the prior multi-view diffusion model. The input of the prior multi-view diffusion model is an image or a text?

L5.  Where is the definition of Personalization? I can’t find any description of the proposed Personalization in Section 4.

L6. In Lines 792-793, the authors claim that PlacidDreamer focuses on the generation of individual objects. However, the PlacidDreamer seems can’t generate multiple objects or objects with surroundings. This is because the prior multi-view diffusion model is trained on Objaverse. This is a drawback of PlacidDreamer.

**Suitability:**

2

---

### Meta-Review · Area_Chair_wnTT · 2024-07-05

**Recommendation:** Accept (Poster)
**Confidence:** 4

**Metareview:**

The paper presents a well-written and primarily clear description of the new PlacidDreamer framework, which focuses on harmonizing initialization, multi-view generation, and text-conditioned generation in 3D asset creation. Key to this approach is the integration of the Latent-Plane module and the novel Balanced Score Distillation (BSD) algorithm, addressing challenges like conflicting optimization directions and over-saturation. The BSD algorithm treats score distillation as a multi-objective optimization problem, resulting in richer details and improved saturation balance in generated 3D assets. The paper also provides extensive qualitative and quantitative evaluations, showcasing the superior performance and effectiveness of the PlacidDreamer framework compared to existing methods.

For the issue raised by reviewer WJz2, the AC has double-checked Table 1 in the rebuttal; the authors have added new baselines from the T3bench and incorporated a new method, LucidDreamer, that employs Gaussian for rendering. For all the comparisons, the approach performs better than the compared approaches.  It appears that the authors also aim to highlight methods that do not require mesh conversion, like LucidDreamer, in Table 1.

Considering all the merits, the AC recommends accepting this paper.  The Authors should clearly address the Reviewers' concerns in the final version.